# Genetic Diversity of Stellate Sturgeon in the Lower Danube River: The Impact of Habitat Contraction upon a Critically Endangered Population

Daniela Nicoleta Holostenco [1,2], Mitică Ciorpac [1,3,*], Elena Taflan [1], Katarina Tošić [1,4], Marian Paraschiv [1], Marian Iani [1], Ștefan Honț [1], Radu Suciu [1] and Geta Rîșnoveanu [2,5,*]

1   Danube Delta National Institute for Research and Development, 820112 Tulcea, Romania;
    daniela.holostenco@ddni.ro (D.N.H.); elena.taflan@ddni.ro (E.T.); katarinatosic@gmail.com (K.T.);
    marian.paraschiv@ddni.ro (M.P.); marian.iani@ddni.ro (M.I.); stefan.hont@ddni.ro (Ș.H.);
    radu.suciu34@gmail.com (R.S.)
2   Doctoral School of Ecology, Faculty of Biology, University of Bucharest, 050095 Bucharest, Romania
3   Genetic Improvement Laboratory, Research Station for Cattle Breeding Dancu, 707252 Iași, Romania
4   Faculty of Biology, University of Belgrade, 11000 Belgrade, Serbia
5   Department of Systems Ecology and Sustainability, Faculty of Biology, University of Bucharest,
    050095 Bucharest, Romania
*   Correspondence: ciorpac.mitica@gmail.com (M.C.); geta.risnoveanu@g.unibuc.ro (G.R.);
    Tel.: +40-765442892 (M.C.); +40-727803028 (G.R.)

**Abstract:** One of the last wild populations of the critically endangered stellate sturgeon (*Acipenser stellatus*) survives in the Danube River. Limited knowledge about the genetic structure, ecology, and evolution of this species led to poor and inconsistent management decisions with an increased risk for species extinction in the wild. Here we show the results of genetic structure screening of the Danube River wild population over 12 years timespan. Our research does not bring evidence of population recovery. No genetic structuring was identified at the mitochondrial level concerning spawning migration timing, sampling locations, and developmental stages. Eleven maternal lineages were revealed based on restriction fragment lengths analysis of the D-loop region, with one haplotype as the most frequent. While this could be the result of a massive restocking activity using a reduced number of spawners, our data does not support it. The selection of mitochondrial haplotypes under the pressure of habitat contraction and the narrower range of temperature variation since dams' construction on the river could explain the observed distribution. Several factors of managerial concern are discussed. Our results provide baseline data on the mtDNA diversity in a critically endangered species of exceptionally high socioeconomic and conservation interest.

**Keywords:** species conservation; wildlife; habitat contraction; anadromous sturgeons; haplotypes; damming; endangered species

## 1. Introduction

In the Black Sea basin still survives the last wild population of stellate sturgeon (*Acipenser stellatus*, Pallas 1771) able to undertake upstream spawning migration into the Danube River [1].

Along with other sturgeon species that spawn in Danube River, during the 20th century, the stellate sturgeon was subject to heavy fishing pressure, leading to a reduction of the population size to critical levels [2]. Completion of the Iron Gates I (in 1972) and II dams (in 1986) halved their historical migration route which used to reach upstream to Bratislava–river kilometer (RKM) 1869 [3], and altered water levels and flow rates, essential factors for the development of different life stages. Additionally, changes in the population structure, water pollution, and illegal fishing contributed to the significant decline of the species stocks. Under these circumstances, the stellate sturgeon in the Black Sea Basin

was listed as endangered species in Appendix II of CITES on 1st of April 1998 [4] and as critically endangered in the Red List of IUCN [1].

Inconsistent regulation measures for sturgeon fishing in the Danube River during 1990–2000 induced an appreciable shifting in the age structure of populations towards the prevalence of young individuals [5] as well as a progressive reduction of the young-of-the-year (YOY) recruitment from natural occurring spawning [6]. Since 1991, several supportive stocking programs were conducted to diminish the extinction risk of the spawning populations for all four remnant sturgeon species (stellate sturgeon–*A. stellatus*, Russian sturgeon–*A. gueldenstaedtii*, sterlet sturgeon–*A. ruthenus* and beluga sturgeon-*Huso huso*). To avoid introgression of exogenous genes in the Danube River gene-pool, wild spawners caught in the Danube River have been used as genetic material in the supportive stocking programs since the beginning [3]. However, these programs alone proved to be insufficient actions for restoring self-sustaining sturgeon populations in the Danube River [7] as no prior investigation regarding the genetic structure of wild populations has been done. So far, a limited number of studies on genetic diversity, population genetic structure, and demographic history of *A. stellatus* have been carried out [8–13], with only a few of them investigating the stellate sturgeon in the Black Sea and Danube River [14–18].

The genetic structure of the stellate sturgeon spawning in the Lower Danube River (LDR) was investigated only in 2008 using nuclear DNA markers [16] followed by Holostenco et al. [18], using D-loop as a mitochondrial marker. None of these studies looks back into the species life history using genetic tools. Moreover, the existence of spring migrants that spawn the same year, autumn migrants that overwinter the river and spawn the next spring [3,13], as well as the different location of spawning grounds along the 863 Km barrier-free reach of the LDR, might be strong ecological indicators of the potential existence of genetic structuring within this population. This kind of knowledge is essential to the development of long-term strategies able to sustainably preserve and restore endangered migratory sturgeon populations.

This paper aims to present the occurrence trends of the maternal lineages, in both adult and young-of-the-year (YOY) stages during a 12-years' time span. Restriction fragment length polymorphism (RFLP) analysis, of the D-loop region was used to: (i) assess the genetic variability within cohorts of adult stellate sturgeons migrating for spawning in the lower Danube River, as well as of YOY on their downstream migration, and (ii) to check whether autumn and spring migration behavior is related to the genetic population structure of migrant adults.

Knowing the genetic structure of a population and the main patterns of its dynamics are key preliminary conditions for a proper management of its conservation and sustainable use. We hypothesized that many decades of unsustainable exploitation and changes in the wild stellate sturgeon population's habitat in the Danube river negatively affected its genetic structure by reducing the diversity.

## 2. Materials and Methods

Over a 12-years timespan (1999–2011), 78 adult stellate sturgeons were captured during upstream migration to the spawning site, in autumn and spring, at different sites along the Sfântu Gheorghe Branch, main Danube River (RKM 100 to 182) and Borcea Branch by scientific fishing (drifting trammel nets) or by commercial fishers (Figure 1, Table 1).

Prior to 2006, all wild adults captured and reported by commercial fishermen along the lower Danube River sector (Figure 1) were sampled. After the sturgeon fishing ban in 2006 adults were captured only for supportive stocking programs using special permits according to Appendix 1 of the joint Ministerial Order 330/262/2006 (republished by Ord. 82/2012) and Ministerial Order 867/2016. YOY specimens (N = 56) were yearly captured and sampled on downstream migration to the sea, during wild sturgeon annual recruitment monitoring at RKM 123, a feeding ground identified in 1997–1999 on the Romanian side of the Danube river, downstream of Reni, Ukraine, where juveniles belonging to different

spawning locations/events upstream the Danube, arrive and feed on their way to the Black Sea. Young fish were captured with bottom drifting trammel nets (95 × 1.8 m, 20 mm mesh size) along 850 m of the river bottom. All captured adults and YOY individuals were released as quickly as possible after the biological material was sampled.

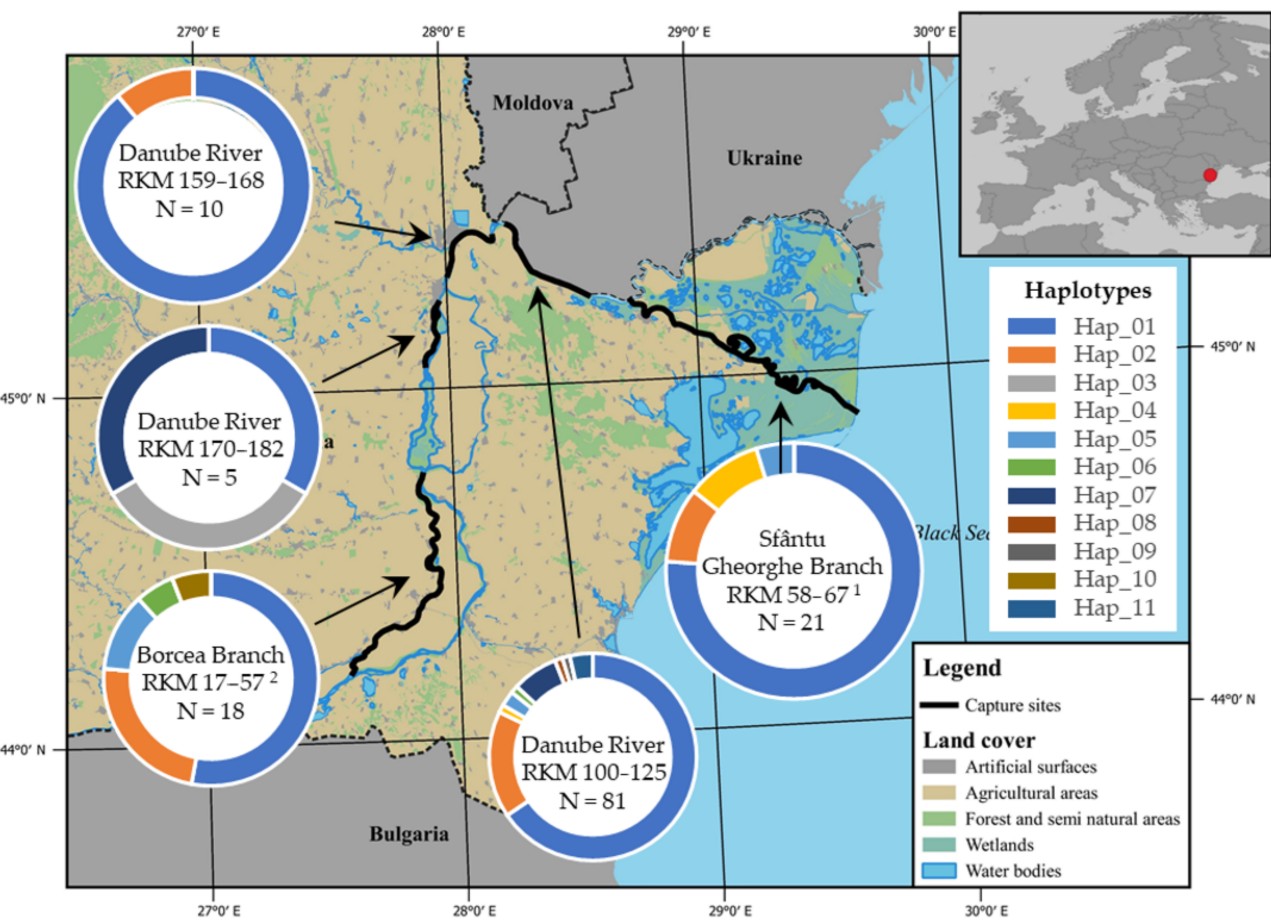

**Figure 1.** Sampling site locations along the lower Danube River and corresponding pie charts showing the frequency distribution of mtDNA composite haplotypes across the study area. ([1] Sfântu Gheorghe Branch has its RKM notation, different from Danube River, from RKM 0 to RKM 110; [2] Borcea Branch has its RKM notation, different from Danube River, from RKM 0 to RKM 100).

**Table 1.** Sampling sites of stellate sturgeon individuals along the lower Danube River, and the number of spring and autumn adult migrants and YOY individuals captured and sampled for genetic analyses per year.

| Season | 1999 | 2000 | 2001 | 2002 * | 2003 * | 2004 * | 2005 | 2006 | 2007 | 2008 | 2009 | 2010 | 2011 | Total |
|---|---|---|---|---|---|---|---|---|---|---|---|---|---|---|
| Upstream migration (adult) | | | | | | | | | | | | | | |
| Spring [a] | 0 | 0 | 0 | 0 | 13 [3] | 5 [3] | 0 | N.A. | 22 [4] | N.A. | N.A. | 11 [5] | 8 [6] | 78 |
| Autumn [b] | 13 [1] | 6 [2] | 0 | 0 | 0 | 0 | 0 | N.A. | N.A. | N.A. | N.A. | N.A. | N.A. | |
| Downstream migration (YOY) | | | | | | | | | | | | | | |
| Summer [c] | 0 | 15 | 0 | 0 | 0 | 0 | 11 | 0 | 0 | 8 | 6 | 16 | 0 | 56 |

*N.A.*–data are not available: fishing activities were not conducted; [a] April–June; [b] September–October; [c] June–August; [1] Sf. Gheorghe Branch RKM: 58–67 = 3 individuals, Danube River RKM: 159–168 = 9 individuals, Danube River RKM: 170–182 = 1 individual; [2] Sf. Gheorghe Branch RKM: 58–67; [3] Danube River RKM: 100–125; [4] Sf. Gheorghe Branch RKM: 58–67 = 12 individuals, Danube River RKM: 100–125 = 10 individuals; [5] Borcea Branch RKM: 17–57; [6] Danube River RKM: 170–182 = 2 individuals, Borcea Branch RKM: 17–57 = 6 individuals; all YOY individuals were captured at a feeding ground in the Danube River RKM 123; * more individuals are reported in the commercial fishing reports, but they were not sampled for the genetic analysis.

Genetic screening of 134 wild individuals was conducted. Small fragments of the anal fin (~1 cm$^2$) collected from live individuals using minimally invasive sampling were stored in 96% ethanol at 4 °C for molecular analysis. Anal fin fragments (~30 mg) were subjected to genomic DNA extraction using the phenol-chloroform protocol according to Taggart et al. [19] following a proteinase K digestion step. The polymerase chain reaction (PCR) was carried out to partially amplify the mitochondrial D-loop region (513 bp) using the CrFs (5′ -GTA GTA AGA GCC GAA CAT CC-3′) and CrRs (5′-GTC CTG CTT TTG GGG TTT GA-3′) primer set and reaction conditions previously described by Onara et al. [20]. Subsequently, corresponding sequences of the mitochondrial D-loop region from stellate sturgeon available in the NCBI GenBank database were aligned and analyzed using BioEdit software [21] to identify potential polymorphic restriction sites. Eight restriction enzymes were used to distinguish the polymorphisms: *Ase*I (NEB R0526), *Rsa*I (Promega R6371), *Hae*III (Promega R6171), *Alu*I (Promega R6281), *Tru9*I (Promega R7011), *Bsr*I (Promega R7241), *Bsl*I (NEB R0555), *Bam*HI (Promega R6021). The restriction fragments were obtained individually for each candidate enzyme.

The RFLP reaction was performed by overnight digestion in a 10 μL final volume, containing 1U of the enzyme, in accordance with manufacturer specifications. Incubation temperature for restriction reactions were as follow: 37 °C for *Ase*I, *Rsa*I, *Hae*III, *Alu*I, *Bam*HI, 65 °C for *Tru9*I, *Bsr*I and 55 °C for *Bsl*I. A 2.5% agarose gel electrophoresis was run to separate the obtained restriction fragments by length. Length quantification was performed by image analysis of the electrophoresis gel visualized in UV light, using a FastRuler Low Range DNA Ladder (Fermentas-SM1103). Additionally, the RFLP patterns from in silico digestions were inferred using the GeneQuest module from the Lasergene package [22]. Fragment patterns were assessed separately, for each enzyme, and further used to construct each haplotype definition. For each sample, the individual enzyme haplotypes were overlapped to form composite haplotypes defined as mixtures of each enzyme's individual haplotypes.

The mtDNA composite haplotypes and their absolute and relative frequencies were used to graphically illustrate occurrence patterns and abundance using MS Excel. The composite haplotype distribution was tested based on a Kolmogorov–Smirnov test simulation (bootstrap 1,000,000 steps) under the normal distribution (Gaussian) fit-distance using fitdistrplus [23] and logspline [24] R [25] libraries. Additionally, two-way PERMANOVA (permutational multivariate analysis of variance) and Kruskal–Wallis test in PAST 4.03, the Palaeontological Association software [26] and Pearson's Chi-squared test (with simulated *p*-value by Monte Carlo simulation, based on 1000 replicates) in R were conducted to test for any genetic structuring (e.g., number of haplotypes and their frequency) related to adult spawning migration season, developmental stages (adults and YOY), year and sampling locations.

## 3. Results

Out of eight restriction enzymes tested, only three showed length polymorphisms for the mitochondrial D-loop region of stellate sturgeon individuals in the lower Danube River. PCR-RFLP screening using *Bsl*I and *Bsr*I restriction enzymes revealed three different haplotypes for each. *Hae*III showed a higher degree of polymorphism exhibiting five different haplotypes (Figure 2). A better resolution of the maternal lineage distribution is reflected by the composite haplotypes constructed by overlapping the individual enzyme haplotypes for certain individuals. Out of the 45 possible composite haplotypes (*Hae*III × *Bsl*I × *Bsr*I = 5 × 3 × 3), only 11 were observed in the lower Danube River stellate sturgeon population (Table 2).

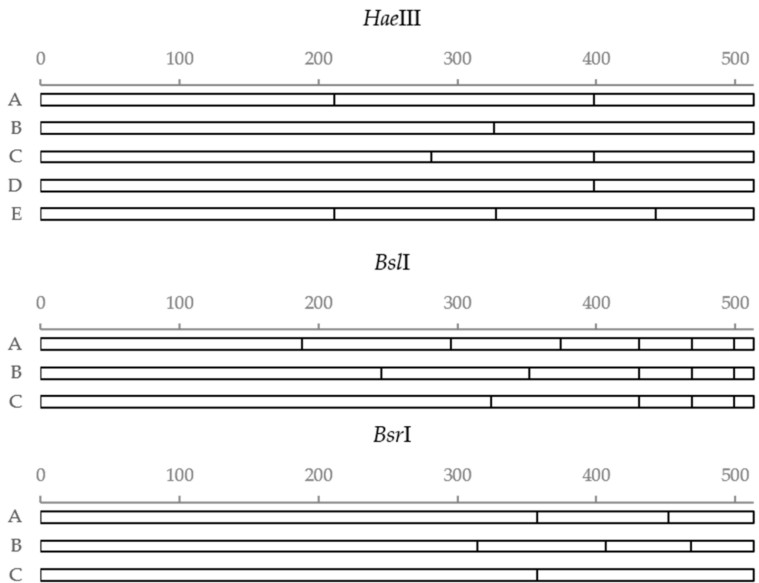

**Figure 2.** Restriction sites distribution of each individual enzyme haplotype.

**Table 2.** The composite haplotypes definition and composition: 11 composite haplotypes were observed out of 45 possible.

| Haplotype Name | Haplotype Definition | Haplotype Composition | | |
|---|---|---|---|---|
| | | *Hae*III | *Bsl*I | *Bsr*I |
| Hap_01 | AAA | A | A | A |
| Hap_02 | ABA | A | B | A |
| Hap_03 | BAA | B | A | A |
| Hap_04 | CAA | C | A | A |
| Hap_05 | AAB | A | A | B |
| Hap_06 | ACA | A | C | A |
| Hap_07 | DAA | D | A | A |
| Hap_08 | BBA | B | B | A |
| Hap_09 | AAC | A | A | C |
| Hap_10 | DBA | D | B | A |
| Hap_11 | EAA | E | A | A |

Based on their relative frequency, five classes of haplotypes were identified. A highly common (frequent) haplotype was h1, exhibited by more than 66% of the individuals and a moderate haplotype, h2, occurred at more than 15% of the samples. The cumulative frequency of these two haplotypes in the metapopulation is 82%. Additionally, one rare haplotype (h7, 5.2%), four very rare haplotypes (h5, h4, h11 and h6, with a cumulative frequency of 9.7%) and four unique haplotypes encountered in only one individual each (h8, h9, h10 and h3 with a cumulative frequency of 2.98%) were observed.

Haplotypes h9 and h11 occurred only in YOY, h8 and h10 were present only in spring migrating adult males, h6 was present in spring migrating adults, whereas h5 and h7 occurred in spring migrating adults and YOY (Table 3). The highest number of haplotypes (9) were recorded on the Danube River at RKM 100–125, being significantly different as compared with the other locations along the Danube River (Kruskal Wallis, $p < 0.05$) except Borcea branch where five haplotypes were present (Figure 1).

**Table 3.** Absolute frequency of haplotypes by migration season, developmental stage and gender.

| Haplotype | Adults | | | | | YOY |
|---|---|---|---|---|---|---|
| | Spring | | Autumn | | | Summer |
| | Female | Male | Female | Male | Unknown | |
| Hap_01 | 16 | 25 | 0 | 4 | 11 | 33 |
| Hap_02 | 2 | 5 | 0 | 1 | 1 | 12 |
| Hap_03 | 0 | 0 | 0 | 0 | 1 | 0 |
| Hap_04 | 0 | 1 | 1 | 0 | 0 | 1 |
| Hap_05 | 1 | 2 | 0 | 0 | 0 | 2 |
| Hap_06 | 1 | 1 | 0 | 0 | 0 | 0 |
| Hap_07 | 1 | 2 | 0 | 0 | 0 | 4 |
| Hap_08 | 0 | 1 | 0 | 0 | 0 | 0 |
| Hap_09 | 0 | 0 | 0 | 0 | 0 | 1 |
| Hap_10 | 0 | 1 | 0 | 0 | 0 | 0 |
| Hap_11 | 0 | 0 | 0 | 0 | 0 | 3 |

The highest number of haplotypes (8) was recorded during the spring and the lowest (4) during the autumn when the total number of captured individuals was also the lowest (18). Seven haplotypes were present in YOY samples (Table 3). One rare haplotype was identified in 1999 and 2004, two in 2003, 2007 and 2010, and three in 2011.

Overall, the haplotypes frequency distribution in the lower Danube River stellate sturgeon population over a 12-years timespan seems to not follow a normal distribution, according to the Kolmogorov–Smirnov test ($p < 0.001$). Based on the frequency distribution of haplotypes (Table 3), no clear patterns (PERMANOVA, $p > 0.05$) in the genetic structure of upstream migrants in relation to season (autumn and spring), nor to sampling location or year, could be distinguished. In contrast, yearly haplotypes frequency distribution showed a different genetic structure (PERMANOVA, $p = 0.0037$) between YOY and adult individuals, independent of the adults sampling location. Moreover, within YOY population, yearly haplotypes frequency distribution pointed out a different genetic pattern for 2005 offspring recruitment (Pearson's Chi-squared, $p = 0.0009$).

## 4. Discussion

Identification of the genetic structure of a population subject to multiple stressors (i.e., unsustainable exploitation for many decades, habitat contraction, and degradation) is a preliminary condition for proper management of its conservation and sustainable use. It is the case of the remaining stellate sturgeon population, an endangered species still naturally spawning in the lower Danube River. Despite its great conservation and economic importance, limited knowledge is available about the genetic structure of this species. Our fishing data demonstrate a high demographic inconsistency of the remnant stellate sturgeon population in the lower Danube River. In seven out of the twelve year timespan, the natural recruitment of the offspring was not quantifiable. Furthermore, during five out of the seven years without YOY capture during the summer downstream migration to the sea, adult individuals were captured during upstream spawning migration. This result likely reflects limited offspring's survival rate in the present habitat and environmental context of the river. Our research revealed a reduced genetic diversity, with the dominance of one composite haplotype out of the 11 haplotypes identified within wild cohorts of adults and YOY. While a certain heterogeneity level was evident in our study, no genetic structuring was identified at the mitochondrial level concerning spawning migration timing, sampling locations, and developmental stages.

### 4.1. Genetic Variability within Cohorts of Adult Stellate Sturgeons Migrating for Spawning in the Lower Danube River and YOY on Their Downstream Migration

Despite the relatively low number of individuals we captured and sampled, primarily due to the reduced remnant population dimension in the lower Danube River, a total

number of 11 composite haplotypes was identified. This is in accordance with the results reported in the literature for other vulnerable sturgeon populations (e.g., 11 and respectively 12 composite haplotypes identified in the case of near-threatened Atlantic sturgeon population [27] and of critical Persian sturgeon) [28]. From the 11 composite haplotypes identified in our study, two haplotypes occurred more frequently than the others (cumulative frequency of >80%). These two frequent haplotypes are well distributed over the entire 12-year time span and in developmental stages, YOY and adults. Genetic studies on mitochondrial markers on other critically endangered sturgeon species also detected one or two haplotypes with high frequency in the population [20,29].

A high frequency of one or two haplotypes in a population could result from a massive restocking activity that uses a reduced number of spawners, in which rare haplotypes represent remnants of the original population, some of which used to spawn upstream the dams and are not able anymore to reach their historic spawning grounds in the actual conditions in the river [30]. According to Iani et al. [31], in the Danube River, the hatchery-produced offspring were tagged in the pectoral fin with coded wire tags only after 2005 [32]. Onără et al. [32] mention that stellate sturgeon first matures at 5–6 years for males, and 7–10 years for females. However, no tagged individuals nor peak of natural recruitment were observed in our samples to demonstrate stocked individuals returned in the river from the sea and affect the genetic structure of the population.

The high frequency of one or two haplotypes in a population may also suggest a natural selection process of the mitochondrial genotype, which most probably appeared as an indirect effect of habitat contraction and was induced by thermal adaptation. Environmental temperature is experienced by mitochondria, and the temperature was suggested as a strong selective factor in maintaining the high homing fidelity of sturgeons (e.g., *Acipenser oxyrinchus desotoi*) [33]. Our study investigated the mitochondrial control region (D-loop), a non-coding region that cannot be under selection. However, due to gene linkage and the lack of recombination between mitochondrial lineages, the D-loop region reflects the entire mitochondrial genome selection. The relative fitness of different genotypes to local conditions will be a selection factor [34]. Bruch and Binkowski [35] proved that water temperature is a critical factor of spawning, triggering the onset and duration of the spawning. When water temperature exceeds the upper or the lower limit, the spawning activity is ceased in females for that year and postponed for males until the right conditions will be met in the same year. Besides the habitat contraction and impact on sturgeon spawning grounds [36] dam construction impact downstream water temperature [37] and homogenize the genetic diversity of sturgeon populations [38]. In the particular case of the Danube River stellate sturgeon, temperature variations across the entire historical distribution range of 1869 Km, upstream to Bratislava may have played a strong role in mtDNA genotype selection. After completing the Iron Gates I and II construction on the Danube river (RKM 942 and 863, respectively), a narrower range of temperature was experienced by the sturgeon populations acting as a limiting spawning factor for the "allochthonous" females having origins in the upper river grounds. Females with genetic makeup from upstream subunits of the metapopulation, which historically used to spawn between slightly different water temperature limits, nowadays should have a narrower temperature range for spawning. Selection of the mtDNA genotypes from locally better-adapted population subunits is likely due to sympatric distribution of the remnant population units. More studies are needed to clarify the involved mechanisms in the Danube River.

Genetic structure (e.g., number of haplotypes and frequency) of adult individuals captured downstream of the Iron Gates dams cannot explain the YOY diversity identified in our research (PERMANOVA, $p = 0.0037$), independent of the adults sampling location and year. A different genetic pattern was observed in the YOY cohort in 2005 (Pearson's Chi-squared, $p = 0.0009$). These results likely reflected a genetic input from reaches located upstream of the Iron Gates dams, resulting from active spawning events of population units that remained blocked in the Middle Danube River after dam construction. The YOY individuals born in the Middle Danube River could follow their instinctual downstream

migration behavior to the sea. The Iron Gates dams should not represent a full barrier for the YOY downstream migration due to their small size, allowing them to pass through the hydropower plant turbines [39]. The sporadic occurrence of stellate sturgeon individuals in the Middle Danube river and its tributaries [30,40] came to support our assumption.

Our results also suggest that the survival rates of juveniles with different genetic structure is not even. Several haplotypes (e.g., h3, h6, h8, h10) were not present in our YOY cohort samples. Onără et al. [32] suggested that hatchery-produced juveniles could not fit for survival in nature. The high demographic inconsistency of the remnant stellate sturgeon population in the lower Danube River revealed by our results, with no quantifiable natural recruitment of the offspring in seven out of 12 years timespan, reflects a limited survival rate of offspring from both wild and hatchery in the present habitat and environmental context of the river. Even in years when the maximum number of haplotypes per year in spring adult migrants was recorded (e.g., 2010, 5 haplotypes), the YOY recruitment could not be quantified.

Even no genetic structuring (e.g., haplotype frequency) within cohorts of adult stellate sturgeons migrating for spawning in the lower Danube River was recorded at the mitochondrial level with respect to sampling location and year, the number of haplotypes (9) recorded on the Danube River at a feeding ground for juveniles (RKM 123) differed significantly (Kruskal Wallis, $p < 0.05$) from that recorded at other locations along the Danube River except Borcea branch known as a spawning ground for stellate sturgeon [41], pointing out the importance of these branch for the species lifecycle.

### 4.2. Genetic Population Structure of Autumn and Spring Migrant Adults

While a certain level of heterogeneity was evident in our study, with the highest numbers of haplotypes (8) being recorded during the spring and six out of nine haplotypes being present either in spring (h5, h6, h7, h8, and h10) or in autumn (h3), it did not match with a significant difference in the frequency distribution of maternal lineages (i.e., composite haplotypes) in relation with the migration behavior, site locations or year. Characterization of the genetic structure in our research was based on only a few autumn adult migrants sampled in 1999 and 2000. It may not have been sufficient to detect subtle genetic differences between spring and autumn migrant adults. Nevertheless, these results suggest that population differentiation caused by differences in spawning timing has to be further investigated if the genetic structure of the natural stocks of stellate sturgeon in the lower Danube River is not to be altered by stocking of hatchery-produced progeny. We cannot say how much heterogeneity at the YOY cohorts' mitochondrial level is due to autumn spawning adult migrants.

Several studies have concluded that conventional PCR-RFLP for the D-loop region of mitochondrial DNA is the most variable mtDNA site (compared to cytochrome b and ND5) and could be a helpful tool for resolving population structure to guide the managerial actions [8,9,17,42–44]. Although our study provides a first overview of the genetic structure diversity of stellate sturgeon inhabiting the Danube River, we likely missed substantial diversity at the nuclear level, which has to stay at the core of further research in this area. Nuclear markers (microsatellite) were proved to be a powerful tool to detect intra- [45–47] and inter-species [48] genetic diversity. However, the genomic DNA quality in some old samples could be a strong limitation [49].

### 4.3. Implications for the Management of Species Conservation

In the past decades, specific wildlife management interventions such as fishing bans and supportive stocking programmes were implemented to recover the stellate sturgeon population listed as a critically endangered species in the Danube River [1]. However, our research does not bring evidence of population recovery. A small catch of adults and the inconsistency of the offspring's natural recruitment during our study period ask for the continuation of supportive stocking programmes and the enforcement and prolongation of

the moratorium on commercial sturgeon fishing in all countries along the Danube River and the Black Sea basin until the population will recover to a self-sustaining level.

The dominance of one or two haplotypes in our samples points out several factors of managerial concern to ensure the actual wild haplotype diversity in the structure of hatchery-produced YOY. Our results demonstrate that if the dominant haplotypes constantly occur over time, the rare haplotypes were present once in several years. This is why we highly recommend designing the supportive stocking programmes for several successive years. Such programs also rely on the identification of the spawning sites and extension of sampling along the upstream dams-free sector of the Danube River. Identification of the Borcea branch adults as the most genetically diverse adults investigated in our study asks for the continuation of the monitoring programme in that location and highlights that this segment of the population could be used as broodstock for the supportive stocking programmes. Spawning sites fidelity of the sturgeon populations in the Danube River are understudied compared to other rivers in the world [30,37–40] and represent a critical gap in knowledge that significantly impede the success of conservation actions and thereby have to be addressed as a high priority in future research.

Despite a not significant difference in the genetic structure of the spring and autumn migrants recorded in our samples, we consider it worth investigating the status of this segment of the population in the wild, and autumn migrating group has to be represented in the Danube River supportive stocking programme.

Our data pointed out the need for the extension of the monitoring of the annual recruitment for YOY in other nursery places identified in the river, collection of more data on offspring's survival rate and monitoring the return of the stocked, tagged, individuals in the Danube and of their participation in the spawning events in the wild as crucial to assess supportive stocking programmes success.

Conservation measures should also address the connectivity of long migrants with their historical spawning grounds upstream of the dams and restoration of degraded essential habitats for sturgeon life stages along the river. Our study also suggests a land-locked / remnant population in the Middle Danube with a possibility for yearlings to drift through the ship locks at Iron Gate dams. A survey upstream and downstream of the dams is needed to investigate this issue.

## 5. Conclusions

This study was performed on critically endangered stellate sturgeon species that still spawn naturally in the Lower Danube River, extremely low-studied despite its socioeconomic and conservative huge importance. Our research is the first that assess the genetic structure and diversity of the population and highlight several important issues for the proper management of species conservation and sustainable use. Even over the 12 years of research, no genetic structuring was identified at the mitochondrial level concerning spawning migration timing, sampling locations and developmental stages in the river's current conditions, our research revealed a reduced genetic diversity, with the dominance of one or two composite haplotypes out the 11 identified within wild cohorts of adults and YOY. While this could be the result of a massive restocking activity using a reduced number of spawners, our data does not support it. The natural selection process of the mitochondrial genotype, which most probably appeared as an indirect effect of habitat contraction and was induced by thermal adaptation, could be involved. Despite a certain level of heterogeneity we revealed, no significant difference in the frequency distribution of maternal lineages in relation to the migration behavior.

The genetic structure (e.g., number of haplotypes and frequency) of adult individuals captured downstream of the Iron Gates dams cannot explain the YOY diversity identified in our research, independent of the adults sampling location and year. These results likely reflected a genetic input resulting from active spawning events of population units that remained blocked in the Middle Danube River after dam construction. We also saw evidence of a limited survival rate of wild and hatchery-born offspring, which impose

managerial measures addressed to enhance the habitat conditions and connectivity in the river and use spawning adults in hatcheries.

This research contributes scientific knowledge that enhances the conservation approach and further develops the general framework for the supportive stocking programs and associated measures to diminish the extinction risk of the remnant spawning stellate sturgeon populations.

**Author Contributions:** Conceptualization, D.N.H., R.S., M.C. and G.R.; methodology, D.N.H., R.S., G.R. and M.C.: validation, D.N.H., G.R. and M.C.; formal analysis, M.C., G.R.; investigation, D.N.H., E.T. and K.T.; resources, E.T., M.P., M.I. and Ș.H.; data curation, D.N.H. and M.C.; writing—original draft preparation, D.N.H., R.S., G.R. and M.C.; writing—review and editing, G.R., M.C., D.N.H., R.S., K.T., E.T., M.P., M.I., Ș.H.; visualization, M.C.; supervision, G.R.; project administration, R.S., M.C.; funding acquisition, R.S., M.C. All authors have read and agreed to the published version of the manuscript.

**Funding:** This research was funded by Innovation Norway and the Romanian Ministry of Environment through the BestCombat project (KnRin/2008/112231) by the Romanian National Authority for Scientific Research (26N/2009 PN 09 26 05 05) and by a grant of the Romanian National Authority for Scientific Research and Innovation CCCDI-UEFISCDI, project number ERANET-COFASP-STURGEoNOMICS within PNCDI III.

**Institutional Review Board Statement:** Not applicable.

**Informed Consent Statement:** Not applicable.

**Data Availability Statement:** The data that support the findings of this study are available from the corresponding author, upon request.

**Acknowledgments:** We thank to Marta Peraita Cachaza for execution of the sampling map.

**Conflicts of Interest:** The authors declare no conflict of interest. The funders had no role in the design of the study; in the collection, analyses, or interpretation of data; in the writing of the manuscript, or in the decision to publish the results.

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
