# Peer review of "Genetic Diversity of Stellate Sturgeon in the Lower Danube River: The Impact of Habitat Contraction upon a Critically Endangered Population"

_water, doi:10.3390/w13081115_

Round 1
Reviewer 1 Report
Holostenco et al. evaluate if genetic variation of Sturgeon populations in the lower Danube varies across age groups, gender and time of the year using the D-loop marker. They did not find any structure and discuss that this might be a consequence of haplotype selection. The outcome of this study seems to have been highly conditioned by the limited information that can be taken from the chose marker system. Since it is likely too late for the authors to produce new data with new markers, I suggest some further analyses and discussion points to enrich the manuscript. These can be found below:
Methods:
The reason why sampling adults in Spring and Fall should be explicitly explained.
Haplotype diversity and number of haplotypes could be used to evaluate if genetic diversity is different between the defined groups.
A haplotype network could add more information about possible genetic structure.
The authors should test other factors that might affect genetic diversity and structure. For example, locality and year. Even if sampling is inconsistent among these variables there might be some interesting trends worth exploring in future studies. Besides these inconsistencies can be discussed.
Results and Discussion
The haplotypes 5 to 8 and 10 seam to have a higher frequency in Spring adult populations. Also, the haplotypes 9 and 11 are only present in YOY. Haplotypes 8 and 10 are only present in males. This may be indicative of some genetic structure. This difference in diversity may reflect some of the mechanism outlined by the authors in the discussion. For example, higher diversity in Spring and YOY may indicates that there is higher diversity coming in than leaving which may be explained by low survival rate of juveniles leading to the exclusion of rare haplotypes.
The main limitation of this study is the lack of information provided by the D-loop RFLP approach which is discussed. The authors state that developing nuclear markers require a high effort and costs, which is not true anymore. With high throughput sequencing technologies, it became possible to produce low coverage shot gun sequencing data that can be used to develop dozens of microsatellite markers. Also, these technologies have been to genotype these markers with higher throughput, lower costs, lower homoplasy, and higher replicability than traditional fragment length analysis. Some of these advances should be mentioned as well and the authors should consider the used of nuclear markers in a more positive way. For more information about the applications of high throughput sequencing for microsatellite discovery and genotyping is available in these publications:
De Barba, M., Miquel, C., Lobréaux, S., Quenette, P. Y., Swenson, J. E., & Taberlet, P. (2017). High‐throughput microsatellite genotyping in ecology: Improved accuracy, efficiency, standardization and success with low‐quantity and degraded DNA. Molecular Ecology Resources, 17(3), 492-507.
Šarhanová, P., Pfanzelt, S., Brandt, R., Himmelbach, A., & Blattner, F. R. (2018). SSR‐seq: Genotyping of microsatellites using next‐generation sequencing reveals higher level of polymorphism as compared to traditional fragment size scoring. Ecology and Evolution, 8(22), 10817-10833.
Tibihika, P. D., Curto, M., Dornstauder-Schrammel, E., Winter, S., Alemayehu, E., Waidbacher, H., & Meimberg, H. (2019). Application of microsatellite genotyping by sequencing (SSR-GBS) to measure genetic diversity of the East African Oreochromis niloticus. Conservation Genetics, 20(2), 357-372.
Curto, M., Winter, S., Seiter, A., Schmid, L., Scheicher, K., Barthel, L. M., ... & Meimberg, H. (2019). Application of a SSR‐GBS marker system on investigation of European Hedgehog species and their hybrid zone dynamics. Ecology and evolution, 9(5), 2814-2832.
Author Response
Please see all our answers in the attachment.

Reviewer 2 Report
The article entitled „Genetic Diversity of Stellate Sturgeon in the Lower Danube River: the Impact of Habitat Contraction upon a Critically Endangered Population”.
The manuscript is well done written and presented. The main aim of the manuscript is very important. The Introduction and the Materials and Methods chapters are clear and comprehensible. The Results chapter well presented (but I have some suggestions) and evaluated. The Discussion chapter is too long, I suggest to abridge it.
Overall, this is a well-written manuscript, very clear and concise. However, there are some minor grammatical error corrections to be considered.
I have some general and specific comments.
Many abbreviations are in the text. Please write full text at first and use the abbreviation after it (e.g. Line 110: PCR).
Figure 1. Please create 1 figure (Map A with correct figure title and description) and two separate tables form Fig. 1.
Figure 2. It is a crazy figure! Please create 4 separate tables (B, C, E, F) and 2 separate figures form Figure 2. The authors can merge the information of E and F parts of Fig 2. and create 1 table from the data.
The reviewer suggest to read and cite these publications:
https://www.mdpi.com/2073-4425/11/7/753
Some little comments:
L3: The Impact
L16 and L24: populations
L17: ecology, and evolution of
L18: with an increased
L31: endangered species; - check whether there is any keywords to be added or else it must end without the semicolon
L42: for the development
L43: pollution, and illegal
L44: Under these circumstances,
L46: Red List – Consider capitalizing the word
L55: the Danube River
L60: genetic structure, and
L61: only a few of them
L75: suggested to use to instead of in order to
L111: Please add information about primer set.
L105: after sampling of the biological material – suggested to be written as - after the biological material was sampled.
L107: 99%??? I think 96% the really value.
L119: of the enzyme
L121: were as follows:
L122: suggested to use to instead of in order to
L133: and to developmental stages – remove to
L172: habitat contraction, and
L192: remove commas - DNA level, since
L220: Azov, and
L273: that the construction of the Ghezhoba dam on the Yangtze River
L282: impedes the success
L283: have to be addressed – change to has to be addressed
L295: of the population
L301: as a critically endangered
Author Response

(The authors gave the same response as above.)

Reviewer 3 Report
The MS 1163053 entitled “Genetic Diversity of Stellate Sturgeon in the Lower Danube River: the Impact of Habitat Contraction upon a Critically Endangered Population” by Holostenco et al. reports the genetic analysis of an endangered sturgeon species. These are original data and surely merits to be published, but some aspects of the MS are confused. Indeed, after reading the MS it is not clear to me which are the main points one can retain. Indeed, the aims of the paper are well cleared out (although likely the marker used is not the more appropriate to gain them), but conclusions are fuzzy and it is not clear in which way the results can help in species management. In other words, it seems that the authors lost their main message along the way. In my opinion, the point authors should face in the discussion is not whether the number of haplotypes observed is comparable to that detected in other species (as reported), but if this number tells something on the temporal or spatial distribution of genetic variability of this species, useful for its management.
I think that revisions are necessary for the figures and text (mainly the discussion). I warmly encourage the authors to do this effort, to better focus which is the contribution of these data to species management.
Find in the attached file my point by point observations

Author Response

(The authors gave the same response as above.)

Round 2
Reviewer 1 Report
The manuscript has improved significantly and all my concerns were addressed. I do not have any further comments.Reviewer 2 Report
Dear Authors,
I accepted all answers and corrections of MS.
Good work!